# Modulation of the Gut Microbiota Alters the Tumour-Suppressive Efficacy of Tim-3 Pathway Blockade in a Bacterial Species- and Host Factor-Dependent Manner

**DOI:** 10.3390/microorganisms8091395

**Published:** 2020-09-11

**Authors:** Bokyoung Lee, Jieun Lee, Min-Yeong Woo, Mi Jin Lee, Ho-Joon Shin, Kyongmin Kim, Sun Park

**Affiliations:** 1Department of Microbiology, Ajou University School of Medicine, Youngtongku Wonchondong San 5, Suwon 442-749, Korea; bokyoung.lee814@gmail.com (B.L.); je0977@naver.com (J.L.); myaliell@naver.com (M.-Y.W.); mijin_lee@hanmail.net (M.J.L.); hjshin@ajou.ac.kr (H.-J.S.); kimkm@ajou.ac.kr (K.K.); 2Department of Biomedical Sciences, The Graduate School, Ajou University, Youngtongku Wonchondong San 5, Suwon 442-749, Korea

**Keywords:** antibiotics, cancer immunotherapy, immune checkpoint inhibitor, gut microbiota

## Abstract

T cell immunoglobulin and mucin domain-containing protein-3 (Tim-3) is an immune checkpoint molecule and a target for anti-cancer therapy. In this study, we examined whether gut microbiota manipulation altered the anti-tumour efficacy of Tim-3 blockade. The gut microbiota of mice was manipulated through the administration of antibiotics and oral gavage of bacteria. Alterations in the gut microbiome were analysed by 16S rRNA gene sequencing. Gut dysbiosis triggered by antibiotics attenuated the anti-tumour efficacy of Tim-3 blockade in both C57BL/6 and BALB/c mice. Anti-tumour efficacy was restored following oral gavage of faecal bacteria even as antibiotic administration continued. In the case of oral gavage of *Enterococcus hirae* or *Lactobacillus johnsonii*, transferred bacterial species and host mouse strain were critical determinants of the anti-tumour efficacy of Tim-3 blockade. Bacterial gavage did not increase the alpha diversity of gut microbiota in antibiotic-treated mice but did alter the microbiome composition, which was associated with the restoration of the anti-tumour efficacy of Tim-3 blockade. Conclusively, our results indicate that gut microbiota modulation may improve the therapeutic efficacy of Tim-3 blockade during concomitant antibiotic treatment. The administered bacterial species and host factors should be considered in order to achieve therapeutically beneficial modulation of the microbiota.

## 1. Introduction

T cell immunoglobulin and mucin domain-containing protein-3 (Tim-3) is an immunoregulatory protein encoded by the Hepatitis A virus cellular receptor 2 (*Havcr2*) gene and is an emerging target for cancer immunotherapy. It was discovered as a molecule that distinguished type 1 helper T (TH1) cells from type 2 helper T (TH2) cells [1]. However, it has also been detected on the surface of exhausted CD8+ T cells and on certain innate cells, including natural killer (NK) cells, monocytes, and dendritic cells [2,3,4]. In T cells, Tim-3 induces inhibitory signalling when its cytoplasmic tail interacts with Lnc-tim3, a long non-coding RNA (lncRNA), instead of HLA-B-associated transcript 3 (Bat3) [5]. Tim-3 is associated with the differentiation of T cells, leading to the formation of effector T cells rather than memory T cells [6]. Tim-3 is also linked to NK cell exhaustion [7]. In addition, Tim-3 impedes the nucleic acid-induced activation of dendritic cells, resulting in the suppression of anti-tumour immunity [8]. Increased cellular expression of Tim-3 and an inverse correlation of Tim-3 expression with cancer prognosis have been reported in various cancers including hepatocellular carcinoma, B cell lymphoma, and pancreatic cancer [9,10,11]. Blocking Tim-3 signalling decreases tumour growth in mouse models [12,13]. Further, the concomitant blockade of Tim-3 and programmed death-1 (PD-1) enhances tumour suppression to a greater extent than blocking either pathway alone [2,14,15,16]. Several mechanisms are believed to underlie the tumour-suppressive effect of Tim-3 blockade, including a decrease in regulatory T cell frequency, functional restoration of tumour-infiltrating T cells, increased dendritic cell recruitment to tumour tissue, and enhanced NK cell activity [7,13,16,17,18,19].

The efficacy of anti-cancer therapies is influenced by gut microbiota composition [20]. Certain enteric microbial enzymes directly modulate the effects of anti-cancer nucleoside analogues. For example, purine nucleoside phosphorylase produced by *Escherichia coli* enhanced fludarabine activity [21]. In contrast, germ-free and antibiotic-treated mice exhibit a diminished response to chemotherapeutic drugs, including cyclophosphamide and oxaliplatin, as their therapeutic activities depend on gut microbiota-associated T cell immune responses and reactive oxygen species (ROS) production [22]. Manipulation of gut microbiota by oral administration of *Enterococcus hirae* enhanced the efficacy of cyclophosphamide in mice [23]. Furthermore, the response to immunotherapy targeting PD-1 or cytotoxic T lymphocyte-associated antigen (CTLA)-4 varies with gut microbiota composition in both tumour-bearing mice and patients [24,25,26]. Microbiota modulation has thus been suggested as a strategy to improve the efficacy of immune checkpoint inhibitors [27]. However, whether microbiota modulation can also enhance immune checkpoint inhibition efficacy in patients receiving antibiotics remains unclear. In particular, the relationship between the efficacy of Tim-3 blockade and gut microbiota composition has not yet been investigated.

In this study, we examined whether gut microbiota modulation influences the efficacy of Tim-3 blockade in mouse tumour models. In particular, the efficacy of gut microbiota modulation for enhancing the tumour-suppressive effects of Tim-3 blockade was evaluated in mice receiving antibiotics. We observed that oral gavage of bacteria altered the gut microbiota of mice despite continuous antibiotic administration. Further, the anti-tumour efficacy of Tim-3 blockade depended on the particular bacterial species administered through oral gavage and the mouse strain.

## 2. Materials and Methods

### 2.1. Cell Culture

B16 melanoma cells (American Type Culture Collection (ATCC), Manassas, VA, USA) and Chinese hamster ovary-K1 (CHO, ATCC) cells were cultured in Dulbecco’s modified Eagle’s medium (DMEM, ThermoFisher, Carlsbad, CA, USA) containing 10% foetal bovine serum (FBS, Carpricorn Scientific, Ebsdorfergrund, Germany), 100 U/mL penicillin, and 100 µg/mL streptomycin (Gibco BRL, Dublin, Ireland). CT-26 BALB/c colon carcinoma cells (provided by Dr. Kwon, Ajou University) [28] were cultured in Roswell Park Memorial Institute (RPMI) 1640 media (GIBCO) containing 10% FBS, 100 U/mL penicillin, and 100 µg/mL streptomycin. HEK-293T cells (provided by Dr. Kwon, Ajou University) were cultured in FreeStyleTM 293 expression medium (Gibco).

### 2.2. Construction of Expression Vectors for Tim-3-Blocking Molecules

The IgV domain of Tim-3 was amplified by PCR using primers (forward primer: 5′-CGG GGT ACC GAT TGG AAA ATG CTT ATG TGT TTG AG and reverse primer: 5′-GAA TTC TGC TTT GAT GTC TAA TTT CAG TTC) and the pIRES2-EGFP-Tim3SVMhIg plasmid [19]. The Tim-3 V-domain-coding DNA segment was inserted into the pSecTag2C vector (ThermoFisher Scientific, Waltham, MA, USA) containing mouse immunoglobulin (mIgG2a) CH2CH3 with and without a hinge region, named pSecTag2C-Tim3VdIg and pSecTag2C-Tim3VmIg, respectively.

### 2.3. Western Blotting for Detection of the Tim3VdIg Protein

CHO cells and HEK-293T cells were transfected with a Tim-3 expression vector using Polyethylenimine (PEI, Polyscience, PA, USA). After 2 days, the culture supernatant was collected and loaded on sodium dodecyl sulfate polyacrylamide gel for electrophoresis (SDS-PAGE) or non-denatured PAGE followed by transfer onto a polyvinylidene difluoride (PVDF) membrane (Bio-Rad, Hercules, CA, USA). The membrane was incubated with an anti-mouse IgG antibody conjugated to horseradish peroxidase (ZYMED^®^ Laboratories, Invitrogen, Carlsbad, CA, USA) and then developed using an enhanced chemiluminescence kit (GE Healthcare, Little Chalfont, UK).

### 2.4. Production and Purification of Tim3VdIg Protein

HEK-293T cells (2 × 10^6^/mL) were transfected with pSecTag2C-Tim-3VdIg using PEI. After 7 days, supernatant was harvested from the culture, and the Tim3VdIg protein was purified using Protein A beads (GE Healthcare, Little Chalfont, UK).

### 2.5. Evaluation of Tumour Growth

Tumour growth experiments in mice were approved by the Institutional Animal Care and Use Committee, Ajou University Medical Center (IACUC protocol #2016-0003). Six-week-old male C57BL/6 (B6) and BALB/c mice were purchased from OrientBio (Gyeonggido, Korea). Mice were maintained in specific-pathogen-free conditions and in separate cages based on strain and treatment. An antibiotic mixture containing 900 mg/L ampicillin (Gold Biotechnology, Saint Louis, MO, USA), 900 mg/L neomycin (BioVision, Milpitas, CA, USA), 900 mg/L metronidazole (BioVision, Milpitas, CA, USA), and 300 mg/L vancomycin (Gold Biotechnology, Saint Louis, MO, USA) was administered to mice via drinking water. Drinking water was replaced every three days. After three weeks, all mice were subcutaneously injected with 100 µL of tumour cells (3 × 10^6^ cells/mL): B16 cells for B6 mice and CT-26 for BALB/c mice. Mice were intraperitoneally injected with Tim-3VdIg (60 µg/mouse) every second day for 12 days after tumour challenge. Tumour progression was assessed every second day by determining tumour volume using the formula: tumour volume = 0.523 × tumour length × (tumour width)^2^.

### 2.6. Oral Administration of Bacteria to Mice

A faecal bacteria stock was prepared by collecting faeces from the large intestine of eight-week-old BALB/c and B6 mice under anaerobic conditions. Faeces were suspended in PBS at a concentration of 60 mg/mL followed by centrifugation at 800× *g* for 3 min. The supernatant was aliquoted for storage at −70 °C until it was administered orally at an amount of 100 µL per mouse, seven times in total, once every third day starting seven days before tumour challenge. *Lactobacillus johnsonii* (Korean Culture Center of Microorganisms) and *Enterococcus hirae* (Korean Culture Center of Microorganisms) were cultured at 37 °C in Brain Heart Infusion media and De Man, Rogosa, and Sharpe (MRS) agar, respectively. The bacteria were cultured to an OD of 1.8 measured at 600 nm (corresponding to 10^9^ CFU/mL) and then aliquoted and cryopreserved in 15% glycerol. Each bacterial suspension (100 µL/head) was administered to a mouse seven times in total, on every third day.

### 2.7. Collection of Gut Microbiota Samples and Bacterial DNA Sequencing

Mice were sacrificed on the eighth day after tumour cell challenge to collect caecal content, which was immediately frozen at −70 °C for microbiome analysis. DNA was extracted from caecal samples using the DNeasyPowerSoil Kit (Qiagen, Hilden, Germany) according to the manufacturer’s instructions. DNA quality was assessed by gel electrophoresis and fluorometry. Sequencing libraries were constructed according to the Illumina 16S Metagenomic Sequencing Library protocols using Herculase II fusion DNA polymerase (Agilent Technologies, Santa Clara, CA, USA) and the universal primer pair specific for the V3–V4 region of the 16S rRNA gene with Illumina adapter overhang sequences. The purified PCR product was quantified according to the qPCR Quantification Protocol Guide (KAPA Library Quantification kits for Illumina Sequencing platforms) and qualified using the TapeStation D1000 ScreenTape (Agilent Technologies, Waldbronn, Germany). Paired-end sequencing was performed with Macrogen (Seoul, Korea) using the MiSeq™ platform (Illumina, San Diego, CA, USA).

### 2.8. Sequence Processing and Taxonomic Assignment

The FLASH (Fast Length Adjustment of SHort reads, 1.2.11) program was employed to merge paired-end reads [29]. Open reference operational taxonomy unit (OTU) picking was utilized using the QIIME-UCLUST and NCBI databases.

### 2.9. Statistics

The statistical significance of differences in tumour growth between groups was analysed using the Student’s *t*-test or ANOVA with Bonferroni correction for multiple comparisons. *p* < 0.05 indicated statistical significance. Significant differences in alpha diversity were computed using the Kruskal–Wallis test with Dunn’s multiple comparison test. Significant differences in beta diversity were computed using PERMANOVA and ANOSIM. Significant differences in dispersion were determined by permDISP.

## 3. Results

### 3.1. Tim-3 V Domain—Mouse IgG Fc Fusion Protein Dimer Exerts Tumour-Suppressive Effects

We previously reported the tumour-suppressive effects of Tim-3 blockade using a Tim-3hIg fusion protein comprised of the Tim-3 variable domain (V) and mucin domain linked to the Fc region of human IgG [19]. The Tim-3 V domain is sufficient to bind to its ligands. Furthermore, Tim-3 dimers may exhibit greater stability while interacting with ligands than that by the Tim-3 monomers. Thus, the Tim3VdIg fusion protein, a dimer of two identical polypeptides consisting of the Tim-3 V domain and mouse IgG hinge and Fc regions, was produced. We first examined the expression of Tim3VdIg in the culture media of CHO cells transformed with the Tim3VdIg expression vector. As Tim3VdIg included the IgG hinge region (Cys-Pro-Pro-Cys-Lys-Cys-Cys-Pro) containing cysteine residues that formed disulfide bonds between two identical Tim3VIg fusion proteins, it was detected as an approximately 110 kDa band in native gels and as a 55 kDa band in its monomeric form in denatured gels (Figure 1a). To clearly show dimer formation, Tim3VmIg lacking the hinge region was compared in parallel. Tim3VdIg and Tim3VmIg proteins were similar in size when denatured but were distinct in native gel. Next, we assessed the purity of Tim3VdIg produced by HEK-293T cells (Figure 1b). Tim3VdIg was detected as a single band in SDS-PAGE. We then assessed the tumour-suppressive effect of purified Tim3VdIg in B6 mice inoculated with B16 melanoma cells (Figure 1c). Mice treated with Tim3VdIg exhibited significantly lower tumour growth when compared with that by control mice injected with PBS (*p* < 0.001).

### 3.2. Oral Administration of Antibiotics to Mice Attenuates the Tumour-Suppressive Effect of Tim-3 Blockade

To investigate the impact of gut microbiota modulation on the tumour-suppressive efficacy of Tim3VdIg, we examined tumour growth in mice administered Tim3VdIg while receiving or not receiving antibiotics, based on a previous report of gut microbiota disturbance by antibiotic treatment [30]. A mixture of ampicillin, neomycin, metronidazole, and vancomycin was administered to mice via drinking water, starting three weeks before tumour challenge, until the end of the experiment. Tumour growth was monitored in B6 and BALB/c mice after injection of B16 melanoma cells and CT-26 colon cancer cells, respectively (Figure 1d–f). Given that immunotherapeutic efficacy may vary with age, we included 8-week-old (Figure 1d) and 1-year-old B6 mice (Figure 1e). Significant tumour suppression by Tim3VdIg treatment was observed in both 8-week-old and 1-year-old B6 mice, as well as in 8-week-old BALB/c mice, compared to controls, starting from day 12 (Figure 1d,f) or 14 (Figure 1e), respectively (*p* < 0.001). However, significant suppression of tumour growth by Tim3VdIg was not observed in mice treated with antibiotics except for in 1-year-old B6 mice on day 14. These results indicate that Tim3VdIg may exert tumour-suppressive effects in subjects with different genetic backgrounds, tumour types, and age, but not in subjects treated with antibiotics.

### 3.3. Oral Administration of Faecal Bacteria or Enterococcus hirae Restores the Tumour-Suppressive Effect of Tim-3 Blockade in Mice Treated with Antibiotics

We next analysed the influence of gut microbiota modulation on the efficacy of Tim3VdIg to ascertain whether the attenuation of the tumour-suppressive effect of Tim3VdIg in mice treated with antibiotics was a result of gut microbiota disturbance (Figure 2). Gut microbiota was modulated by feeding antibiotic-treated mice with faecal bacteria prepared from normal mouse faeces, *Enterococcus hirae*, or *Lactobacillus johnsonii* once every three days. While the tumour-suppressive effect of Tim3VdIg was consistently suppressed in antibiotic-treated mice, it was partially but significantly restored by the oral transfer of faecal bacteria or *E. hirae* (Figure 2a). In the case of B6 mice fed with *L. johnsonii*, tumour suppression was partially restored on days 8 and 10 (the suppression percentage was approximately 70% and 60% on days 8 and 10, respectively). However, tumour suppression was not maintained thereafter (the suppression percentage declined to the level of the antibiotic-treated group without oral gavage) (Figure 2b). In BALB/c mice, the tumour-suppressive effect of Tim3VdIg was partially restored after oral gavage of faecal bacteria, but not of *E. hirae* or *L. johnsonii* (Figure 2c,d). These results indicate that gut microbiota modulation may enhance the tumour-suppressive effects of Tim-3 blockade.

To verify changes in the gut microbiota of mice receiving antibiotics and bacteria, we analysed the microbiome of the cecum harvested on the eighth day after tumour challenge. At that time point, antibiotic treatment was ongoing, and oral gavage of bacteria had been performed five times in total. Read count ranged from 69,012 to 106,016 per sample. Rarefaction measurement of 12,311 reads per sample revealed a sufficiently covered diverse microbiome (Figure 3a). We compared the alpha diversity, which represents the complexity of the microbiome within a sample, using the Chao1 and Shannon methods (Figure 3b–e). As expected, alpha diversity was higher in groups not treated with antibiotics compared to groups receiving antibiotics, although a significant difference was not observed in all comparisons. This may be due to the small sample size. Although oral gavage restored the tumour-suppressive efficacy of Tim3VdIg in antibiotic-treated mice, it did not significantly increase alpha diversity.

We then analysed beta diversity using Bray–Curtis distance (Figure 4). Principal coordinate analysis revealed the differences between samples from mice treated with Tim3VdIg (indicated as B_T for B6 and C_T for BALB/c) and samples from mice treated with both Tim3VdIg and antibiotics (indicated as B_T/A for B6 and C_T/A for BALB/c mice) (*R* = 0.813, *p* = 0.001 via ANOSIM). B_T and B_T/A were separated along PC2, whereas C_T and C_T/A were separated along PC1. The B6 groups administered Tim3VdIg, antibiotics, and bacteria (indicated as B_T/A/F for transfer of faecal bacteria and B_T/A/E for transfer of *E. hirae*) were separated from B_T and B_T/A along PC1 as well as along PC2. The samples from BALB/c mice treated with Tim3VdIg, antibiotics, and faecal bacteria (indicated as C_T/A/F) were separated from C_T and C_T/A along PC1, as well as from the samples from mice treated with Tim3VdIg, antibiotics, and *E. hirae* (indicated as C_T/A/E) along PC1 and PC2. Notably, B_T/A/F, B_T/A/E, and C_T/A/F, the groups exhibiting restoration of Tim-3VdIg anti-tumour efficacy, were clustered. Furthermore, C_T/A/E, in which tumour suppression was not restored, was proximal to B_T/A. These results demonstrated clustering of experimental groups based on tumour suppression.

We then compared microbiome composition at the phylum, class, and order levels (Figure 5). A major bacterial population belonging to *Firmicutes* (phylum) *Clostridia* (class) *Clostridiales* (order) was observed in the B_T group, whereas in the C_T group, two major populations belonging to *Bacteroidetes* (phylum) *Bacteroidia* (class) *Bacteroidales* (order) and *Firmicutes Clostridia Clostridiales*, respectively, were observed. In the B_T/A group of B6 mice and in all groups of BALB/c mice (C_T/A, C_T/A/F, and C_T/A/E), *Proteobacteria Gammaproteobacteria Enterobacterales* were predominant (except for C_T/A, in which *Verrucomicrobia Verrucomicrobiae Verrucomicrobiales* were most frequent). In B_T/A/F and B_T/A/E, there were two dominant populations, namely, *Proteobacteria Gammaproteobacteria Enterobacterales* and *Firmicutes Bacilli Lactobacillales*. These results indicated that antibiotic treatment drastically altered the microbiome, and feeding antibiotic-treated mice with bacteria did not fully reconstitute gut microbiota even though it restored the tumour-suppressive effect of Tim3VdIg.

We then analysed differences in microbiota composition at the species level between mouse groups of the same strain in order to identify species associated with the effect of microbiota modulation on Tim3VdIg efficacy (Table 1 and Table 2). The abundance of 45 and 41 bacterial species significantly varied in B6 and BALB/c groups, respectively. Two species, namely, *Beduini massiliensis* and *Propionispira paucivorans*, were potentially associated with the suppression of anti-tumour efficacy of Tim3VdIg in B6 mice, as they were observed in the T/A group but not in the T, T/A/F, and T/A/E groups. The increased abundance of *Proteus alimentorum* and *Akkermansia muciniphila* in T/A/E or T/A/F relative to T/A was indicative of their potential role in enhancing the anti-tumour effects of Tim3VdIg in B6 mice. We could not identify any bacterial species whose modulation may be associated with the tumour-suppressive efficacy of Tim3VdIg in BALB/c mice. Finally, we compared the microbiota composition of Tim3VdIg-treated B6 and BALB/c mice, as oral gavage of *E. hirae* enhanced the tumour-suppressive efficacy of Tim3VdIg in B6 mice but not in BALB/c mice (Table 3). Among the 117 bacterial species identified, 10 species were significantly more abundant in B6 mice, and 16 species were more abundant in BALB/c mice, while 91 species were similarly abundant between these mouse strains. Taken together, these results demonstrate that gut microbiota varies with mouse strain, antibiotic treatment, and oral transfer of bacteria, as well as that gut microbiota modulation affects the tumour-suppressive efficacy of Tim-3 blockade.

## 4. Discussion

Tim-3 pathway blockade is a promising form of cancer immunotherapy. Thus, the identification of factors that affect its anti-tumour efficacy is of great significance. In the current study, the anti-tumour effect of Tim3VdIg, a novel Tim-3-blocking molecule, was demonstrated, and the influence of gut microbiota on the therapeutic effects of Tim3VdIg in mice was analysed. Firstly, the anti-tumour effect of Tim3VdIg was attenuated in mice with antibiotic-elicited gut dysbiosis. Secondly, oral gavage of bacteria restored the anti-tumour efficacy of Tim3VdIg in antibiotic-treated mice, even though their gut microbiota composition was distinct from that of mice not treated with antibiotics. Thirdly, the restorative effect of bacterial transfer on the anti-tumour efficacy of Tim3VdIg varied based on the bacterial species administered and the mouse strain.

The anti-tumour effect of Tim-3 blockade has been reported using anti-Tim-3 antibodies or the monomeric form of the Tim-3-Ig fusion protein [14,19]. The Tim3VdIg used in the current study is a dimer form of the Tim-3V domain and Ig fusion protein and is not commercially available. While it was expected that Tim3VdIg may bind to Tim-3 ligands more stably than that by the Tim3VmIg monomer, the tumour-suppressive effect of Tim3VdIg was comparable to that of Tim3VmIg. Tim3VdIg injection inhibited tumour growth in both C57BL/6 and BALB/c mouse strains previously challenged with melanoma and colon carcinoma cells, respectively, indicative of its anti-tumour effect in hosts with different genetic backgrounds and different tumour types.

Tim-3 is an immune checkpoint molecule, as are PD-1 and CTLA-4. These proteins are targets of cancer immunotherapy. In line with our results, the impact of gut microbiota on the efficacy of PD-1 and CTLA-4 blockade has been previously reported [24,25]. In both mice and humans, the response to PD-1 and CTLA-4 blockade has been associated with gut microbiota composition [25,31]. The presence of *Bifidobacterium* and *Bacteroides* species in mouse gut microbiota has been correlated with tumour suppression by PD-1 and CTLA-4 blockade, respectively [31,32]. In patients with non-small cell lung carcinoma or renal cell carcinoma, clinical responsiveness to PD-1 blockade was associated with *Akkermansia muciniphila* and the T cell response against this species [33]. Notably, *A. muciniphila* was also positively correlated with the tumour-suppressive effect of Tim3VdIg in B6 mice in our study (Table 1). Along with *A. muciniphila*, *Proteus alimentorum* was also associated with an improved response to Tim3VdIg in B6 mice (Table 1). However, our findings were limited by the low number of BALB/c mice samples available for microbiome analysis. Thus, no bacterial species was found to be significantly correlated with the tumour-suppressive effect of Tim-3VdI in this strain. Nevertheless, the current results clearly demonstrate the influence of gut microbiota on the therapeutic efficacy of Tim-3 blockade.

Although mice were continuously treated with antibiotics, oral gavage of bacteria successfully restored Tim3VdIg efficacy. Various studies have reported that the administration of antibiotics causes transient gut dysbiosis, the extent of which depends on the duration and frequency of antibiotic administration, as well as on the bacterial coverage of the antibiotics administered [30,34]. However, it was difficult to find a previous report on the recovery of gut microbiota through concomitant oral administration of bacteria during the course of antibiotic treatment. Cancer patients may have to take antibiotics before or during immunotherapy as treatment against infections. Antibiotics may also be administered prophylactically prior to medical procedures such as cystoscopy in bladder cancer patients, which is performed in order to monitor tumour recurrence. We evaluated the influence of the oral administration of bacteria on Tim3VdIg efficacy in mice receiving antibiotics and observed a favourable therapeutic effect. Although neither gut microbiota diversity nor composition were fully recovered by oral bacterial administration (Figure 3 and Figure 4), alterations in gut microbiota composition were observed (Figure 4 and Figure 5). Furthermore, clustering of mouse groups according to tumour suppression efficacy and oral bacterial administration through principal coordinate analysis of the microbiome suggested that the presence or absence of specific bacterial species may affect the efficacy of Tim3VdIg to a greater extent than that by microbiome alpha diversity. A recent study reported that 28% of cancer patients take antibiotics within 60 days before or 30 days after their first treatment with a PD-1 inhibitor. Further, treatment outcomes observed in these patients were poor compared to those in patients who did not take antibiotics [33]. Our results indicate that modulation of the gut microbiota during anti-tumour immunotherapy may be beneficial in such cases.

Recovery of Tim3VdIg efficacy through the oral administration of bacteria depended on the transferred bacterial species and the host strain. The transfer of *E. hirae*, but not *L. johnsonii*, stably upregulated the efficacy of Tim-3 blockade in B6 mice. Similar to our results, *E. hirae*, but not *L. johnsonii*, improved the efficacy of cyclophosphamide in B6 mice pretreated with antibiotics [23]. However, it should be noted that *L. johnsonii* is absent in mice that are more susceptible to cancers. Further, this species can activate NK cells and lower the kynurenine–tryptophan ratio, which is associated with immune suppression in healthy humans [35,36,37]. *E. hirae* crosses the small intestinal epithelial barrier, migrates to peripheral lymphoid organs, and upregulates the cytotoxic/regulatory T cell ratio in tumour tissues [23]. Additionally, lipoteichoic acid from *E. hirae* elicits the release of several cytokines, including tumour necrosis factor-α, in mice [38]. Further, *E. hirae*-specific T cell responses are correlated with favourable outcomes in cancer patients [39]. Contrary to data obtained from B6 mice, BALB/c mice did not benefit from administration of *E. hirae*. Notably, administration of their own faecal bacteria induced a greater restoration of the anti-tumour efficacy of Tim3VdIg in B6 mice than in BALB/c mice (70–80% vs 40–50% suppression). The cause of this difference remains unknown. It may be attributed to variations in the intestinal response to transferred bacteria, differences in the immune response, and differences in gut microbiota composition between B6 and BALB/c mice. In agreement with this possibility, differences in intestinal gene expression between the BALB/c and C57BL/6 mice after probiotic treatment have been reported [40]. Furthermore, studies have demonstrated a propensity towards a TH1- and TH2-dominant response in B6 and BALB/c mice, respectively [41,42]. Our study also revealed differences in the gut microbiome between B6 and BALB/c mice (Table 3). Such host factors may interact to determine the impact of gut microbiota modulation on the tumour-suppressive activity of Tim-3 blockade.

## 5. Conclusions

Our results emphasize the critical role of gut microbiota in cancer immunotherapy via Tim-3 blockade and reveal the beneficial effect of gut microbiota modulation on tumour suppression even during the continuous administration of antibiotics. Further research will help ascertain the appropriate modulation of gut microbiota via the selection of appropriate bacteria and the careful consideration of relevant host factors.

## Figures and Tables

**Figure 1 microorganisms-08-01395-f001:**
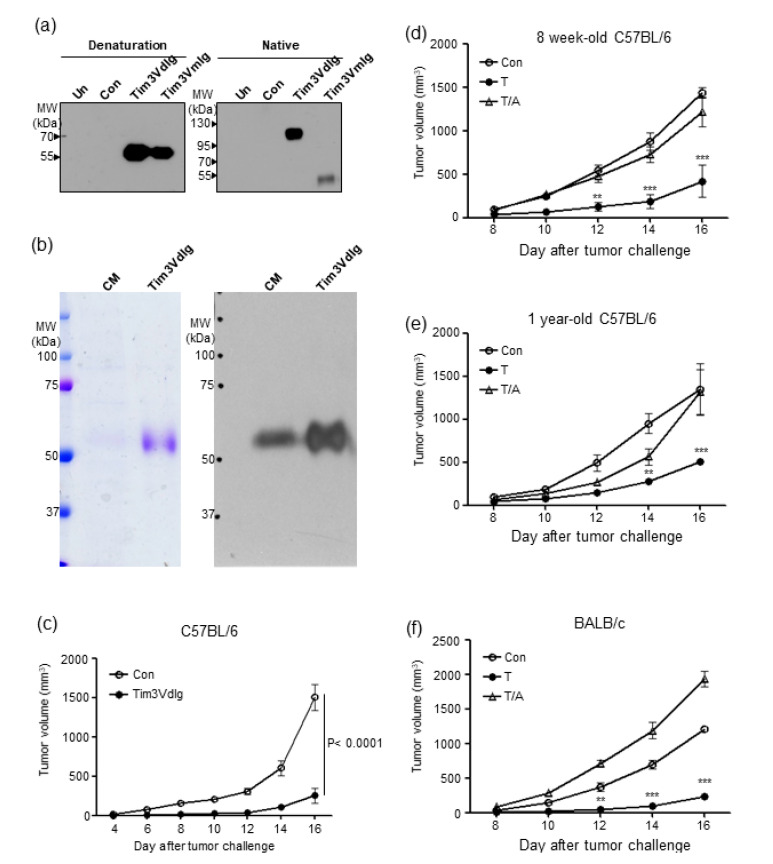
Antibiotic administration hinders the tumour-suppressive effect of Tim3VdIg in two different tumour mouse models. (**a**) Tim3VdIg is expressed as a dimer linked via disulfide bonds between identical polypeptides consisting of the Tim-3 V domain and mouse IgG2a hinge-Fc domain. The culture supernatants of non-transfected CHO cells (Non) and CHO cells transfected with pSecTag2C (Con), pSecTag2C-Tim3VdIg, or pSecTag2C-Tim3VmIg were analysed using Western blotting in both native and denatured conditions. Tim3VmIg lacks the IgG2a hinge region. (**b**) Tim3VdIg purified from culture media of transfected HEK-293T cells was examined using SDS-PAGE (Left) and Western blotting (Right). (**c**) Tumour growth in mice injected with PBS (Con, *n* = 4) or Tim3VdIg (60 µg/mouse, *n* = 4) five times, once every second day after B16 melanoma cell challenge (3 × 10^5^). Tumour growth in 8-week- (**d**) or 1-year- (**e**) old B6 mice injected with B16 cells (3 × 10^5^). Tumour growth in 8-week-old BALB/c mice (**f**) injected with CT-26 cells (3 × 10^5^). Tim3VdIg was injected five times, once every second day after tumour challenge, in mice of two groups. One group received orally administered antibiotics (T/A), and the other did not (T). Control group (Con) mice were treated with PBS. Data are presented as mean ± standard deviation. (*n* = 3 to 8 per group). ** *p* < 0.01, *** *p* < 0.001 vs Con.

**Figure 2 microorganisms-08-01395-f002:**
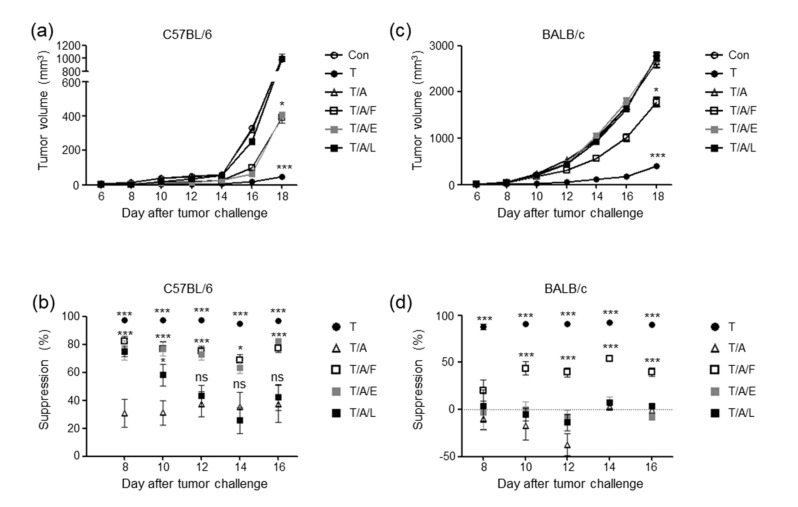
Oral gavage of bacteria restores Tim3VdIg efficacy in antibiotic-treated mice depending on the bacterial species administered and the mouse strain. Tumour growth in B6 (**a**) and BALB/c (**c**) mice injected with B16 melanoma cells and CT-26 cells, respectively. Control group (Con, 8 B6 mice and 4 BALB/c mice) mice were injected with PBS, and the experimental groups were injected with Tim3VdIg once every second day after tumour challenge. Antibiotics were administered via the drinking water. Bacteria were administered to mice through oral gavage seven times in total, once every third day starting seven days before tumour challenge. T: Tim3VdIg treatment alone (9 B6 mice and 10 BALB/c mice). T/A: Tim3VdIg and antibiotic treatment (10 B6 mice and 9 BALB/c mice). T/A/F: Tim3VdIg, antibiotics, and faecal bacteria (10 B6 mice and 9 BALB/c mice). T/A/E: Tim3VdIg, antibiotics, and *Enterococcus hirae* (7 B6 mice and 6 BALB/c mice). T/A/L: Tim3VdIg, antibiotics, and *Lactobacillus johnsonii* (7 B6 mice and 6 BALB/c mice). Tumour suppression (**b**,**d**) was calculated as (1−tumour volume of each mouse relative to the mean tumour volume of control group mice) × 100. Data represent the mean ± standard deviation of two independent experiments. * *p* < 0.05, *** *p* < 0.001 vs Con or T/A.

**Figure 3 microorganisms-08-01395-f003:**
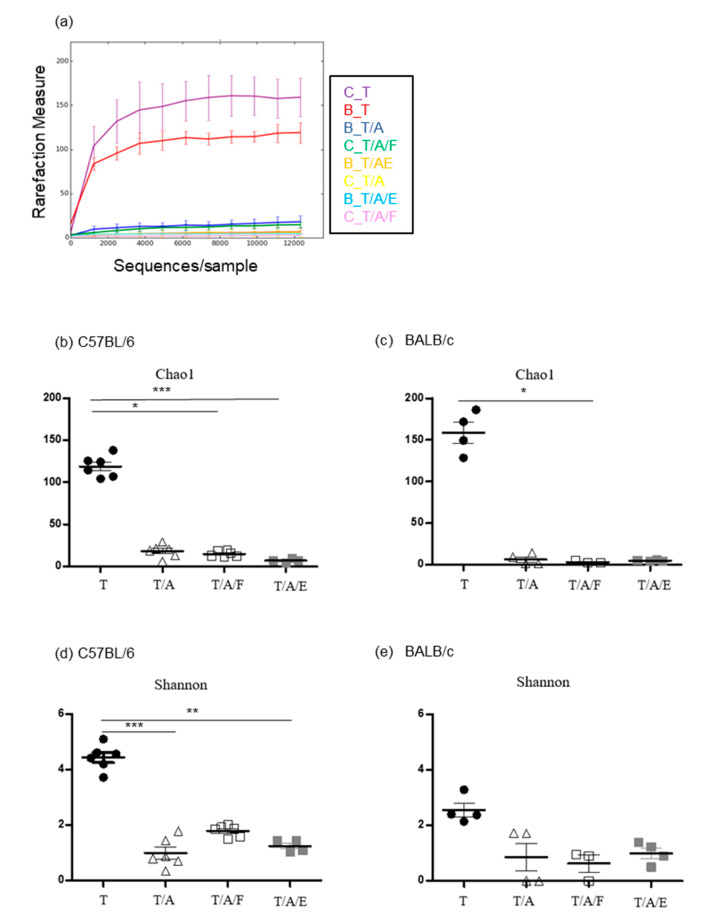
Oral gavage of bacteria does not increase microbiome alpha diversity in antibiotic-treated mice. The microbiome was analysed by 16S rDNA V3V4 sequencing using caecal content obtained from mice on the 8th day after tumour challenge. (**a**) Rarefaction curves after adjusting the read number of each sample by subsampling. (**b**–**e**) Alpha diversity determined by the Chao1 or Shannon method. Each symbol represents each sample. C_T and B_T: BALB/c and B6 mice treated with Tim3VdIg alone, respectively; C_T/A and B_T/A: BALB/c and B6 mice treated with Tim3VdIg and antibiotics; C_T/A/F and B_T/A/F: BALB/c and B6 mice treated with Tim3VdIg, antibiotics, and faecal bacteria; C_T/A/E and B__T/A/E: BALB/c and B6 mice treated with Tim3VdIg, antibiotics, and *Enterococcus hirae*. Data represent mean ± standard deviation. * *p* < 0.05, ** *p* < 0.01, *** *p* < 0.001.

**Figure 4 microorganisms-08-01395-f004:**
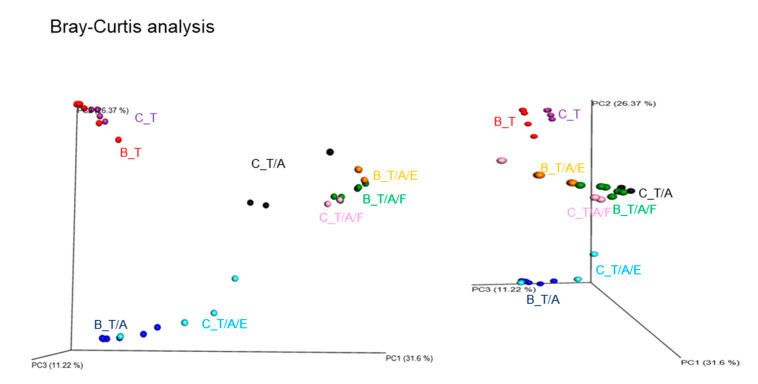
Beta diversity analysis separates samples based on treatment type and mouse strain. Principal coordinate analysis of microbiome samples using Bray–Curtis distance. The microbiome of caecal content obtained from mice on the 8th day after tumour challenge and the indicated treatment was analysed. Each symbol represents each sample. C_T and B_T: BALB/c and B6 mice treated with Tim3VdIg alone, respectively; C_T/A and B_T/A: BALB/c and B6 mice treated with Tim3VdIg and antibiotics; C_T/A/F and B_T/A/F: BALB/c and B6 mice treated with Tim3VdIg, antibiotics, and faecal bacteria; C_T/A/E and B__T/A/E: BALB/c and B6 mice treated with Tim3VdIg, antibiotics, and *Enterococcus hirae*. *p* = 0.001 via PERMANOVA, *p* = 0.013 via permDISP, *R* = 0.813, *p* = 0.001 via ANOSIM.

**Figure 5 microorganisms-08-01395-f005:**
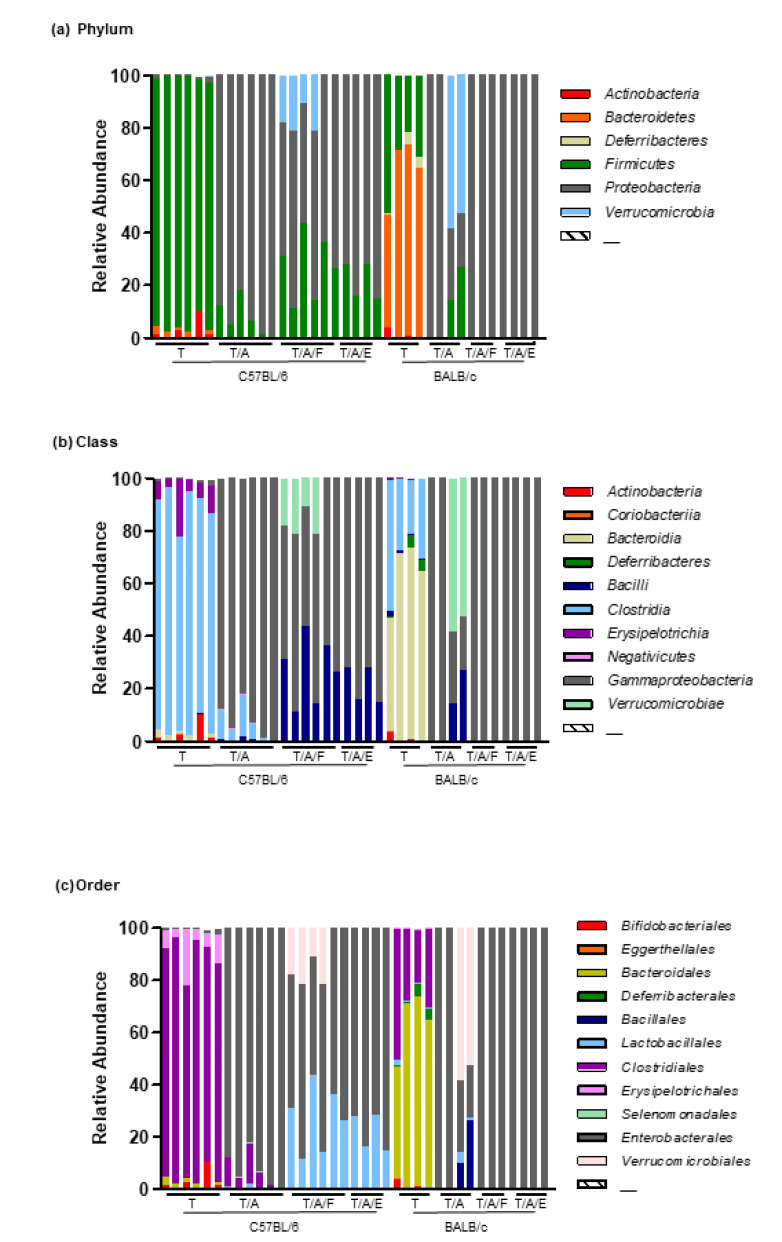
Relative bacterial abundance across mouse strains and treatments. Relative bacterial abundance at the phylum level (**a**), class level (**b**), and order level (**c**). Each bar represents each sample. T: treatment with Tim3VdIg alone; T/A: treatment with Tim3VdIg and antibiotics; T/A/F: treatment with Tim3VdIg, antibiotics, and faecal bacteria; T/A/E: treatment with Tim3VdIg, antibiotics, and *Enterococcus hirae*.

**Table 1 microorganisms-08-01395-t001:** Bacterial species with significantly varied abundance between C57BL/6 groups based on treatment ^1^.

		*p* Value ^2^						
Phylum	Species		T vs T/A	T vs T/A/F	T vs T/A/E	T/A vs T/A/F	T/A vs T/A/E	T/A/F vs T/A/E
*Actinobacteria*	*Asaccharobacter celatus*	0.0011	0.0052	0.0052	0.0116	1.0000	1.0000	1.0000
*Actinobacteria*	*Bifidobacterium pseudolongum*	0.0001	0.0011	0.0011	0.0033	1.0000	1.0000	1.0000
*Bacteroidetes*	*Bacteroides xylanolyticus*	0.0003	0.0045	0.0012	0.0045	1.0000	1.0000	1.0000
*Firmicutes*	*Acetatifactor muris*	0.0001	0.0011	0.0011	0.0033	1.0000	1.0000	1.0000
*Firmicutes*	*Anaerocolumna cellulosilytica*	0.0065	0.0219	0.0219	0.0373	1.0000	1.0000	1.0000
*Firmicutes*	*Anaerocolumna jejuensis*	0.0001	0.0011	0.0011	0.0033	1.0000	1.0000	1.0000
*Firmicutes*	*Anaerotaenia torta*	0.0001	0.0011	0.0011	0.0033	1.0000	1.0000	1.0000
*Firmicutes*	*Beduini massiliensis*	0.0011	0.0052	1.0000	1.0000	0.0052	0.0116	1.0000
*Firmicutes*	*Blautia producta*	0.0007	0.0578	0.0013	0.0048	0.6363	0.6363	1.0000
*Firmicutes*	*Caecibacterium sporoformans*	0.0001	0.0011	0.0011	0.0033	1.0000	1.0000	1.0000
*Firmicutes*	*Christensenella massiliensis*	0.0006	1.0000	0.0114	0.0218	0.0114	0.0218	1.0000
*Firmicutes*	*Clostridium aerotolerans*	0.0011	0.0052	0.0052	0.0116	1.0000	1.0000	1.0000
*Firmicutes*	*Clostridium asparagiforme*	0.0001	0.0011	0.0011	0.0033	1.0000	1.0000	1.0000
*Firmicutes*	*Clostridium Cocleatum*	0.0001	0.0011	0.0011	0.0033	1.0000	1.0000	1.0000
*Firmicutes*	*Clostridium indolis*	0.0013	0.8387	0.054	0.0814	0.0063	0.0169	1.0000
*Firmicutes*	*Clostridium methylpentosum*	0.0065	0.0219	0.0219	0.0373	1.0000	1.0000	1.0000
*Firmicutes*	*Clostridium saccharolyticum*	0.0001	0.0011	0.0011	0.0033	1.0000	1.0000	1.0000
*Firmicutes*	*Desulfitobacterium metallireducens*	0.0001	0.0011	0.0011	0.0033	1.0000	1.0000	1.0000
*Firmicutes*	*Eubacterium coprostanoligenes*	0.0001	0.0011	0.0011	0.0033	1.0000	1.0000	1.0000
*Firmicutes*	*Falcatimonas natans*	0.0065	0.0219	0.0219	0.0373	1.0000	1.0000	1.0000
*Firmicutes*	*Flintibacter butyricus*	0.0001	0.0011	0.0011	0.0033	1.0000	1.0000	1.0000
*Firmicutes*	*Hungateiclostridium thermocellum*	0.0001	0.0011	0.0011	0.0033	1.0000	1.0000	1.0000
*Firmicutes*	*Hydrogenoanaerobacterium saccharovorans*	0.0001	0.0011	0.0011	0.0033	1.0000	1.0000	1.0000
*Firmicutes*	*Kineothri alysoides*	0.0003	0.0012	0.0045	0.0045	1.0000	1.0000	1.0000
*Firmicutes*	*Lachnoclostridium pacaense*	0.0001	0.0011	0.0011	0.0033	1.0000	1.0000	1.0000
*Firmicutes*	*Lactobacillus animalis*	0.0004	0.3507	0.0011	0.0081	0.0666	0.1573	0.8399
*Firmicutes*	*Muricomes intestini*	0.0001	0.0011	0.0011	0.0033	1.0000	1.0000	1.0000
*Firmicutes*	*Murimonas intestini*	0.0001	0.0011	0.0011	0.0033	1.0000	1.0000	1.0000
*Firmicutes*	*Neglecta timonensis*	0.0001	0.0011	0.0011	0.0033	1.0000	1.0000	1.0000
*Firmicutes*	*Oscillibacter ruminantium*	0.0001	0.0011	0.0011	0.0033	1.0000	1.0000	1.0000
*Firmicutes*	*Oscillibacter valericigenes*	0.0001	0.0011	0.0011	0.0033	1.0000	1.0000	1.0000
*Firmicutes*	*Phocea massiliensis*	0.0001	0.0011	0.0011	0.0033	1.0000	1.0000	1.0000
*Firmicutes*	*Propionispira paucivorans*	0.0065	0.0219	1.0000	1.0000	0.0219	0.0373	1.0000
*Firmicutes*	*Pseudobutyrivibrio ruminis*	0.0001	0.0011	0.0011	0.0033	1.0000	1.0000	1.0000
*Firmicutes*	*Pseudoflavonifractor capillosus*	0.0065	0.0219	0.0219	0.0373	1.0000	1.0000	1.0000
*Firmicutes*	*Pseudoflavonifractor phocaeensis*	0.0001	0.0011	0.0011	0.0033	1.0000	1.0000	1.0000
*Firmicutes*	*Roseburia hominis*	0.0001	0.0011	0.0011	0.0033	1.0000	1.0000	1.0000
*Firmicutes*	*Roseburia intestinalis*	0.0011	0.0052	0.0052	0.0116	1.0000	1.0000	1.0000
*Firmicutes*	*Ruminococcus gnavus*	0.0003	0.0045	0.0012	0.0045	1.0000	1.0000	1.0000
*Firmicutes*	*Ruthenibacterium lactatiformans*	0.0003	0.0045	0.0012	0.0045	1.0000	1.0000	1.0000
*Firmicutes*	*Turicibacter sanguinis*	0.0008	0.2083	0.0016	0.0056	0.2675	0.2675	1.0000
*Proteobacteria*	*Escherichia fergusonii*	0.0003	0.2236	0.0001	0.0669	0.0627	0.4252	0.3023
*Proteobacteria*	*Klebsiella variicola*	0.0002	0.1948	0.1367	0.1367	0.001	0.0017	0.8369
*Proteobacteria*	*Proteus alimentorum*	0.0002	0.3285	0.0069	0.0003	0.3285	0.0347	0.3285
*Verrucomicrobia*	*Akkermansia muciniphila*	0.0014	0.9065	0.0531	0.9065	0.0027	1.0000	0.0085

^1^ Mouse groups: T (Tim3VdIg-treated group), T/A (Tim3VdIg- and antibiotic-treated group), T/A/F (Tim3VdIg-, antibiotic-, and faecal bacteria-treated group), T/A/E (Tim3VdIg-, antibiotic-, and *E. hirae*-treated group).^2^ Analysed by the Kruskal–Wallis test with Dunn’s multiple comparison.

**Table 2 microorganisms-08-01395-t002:** Bacterial species with significantly varied abundance between BALB/c groups based on treatment ^1^.

			*p* Value ^2^					
Phylum	Species		T vs T/A	T vs T/A/F	T vs T/A/E	T/A vs T/A/F	T/A vs T/A/E	T/A/F vs T/A/E
*Actinobacteria*	*Bifidobacterium pseudolongum*	0.004	0.014	0.019	0.014	1.000	1.000	1.000
*Bacteroidetes*	*Bacteroides caccae*	0.004	0.014	0.019	0.014	1.000	1.000	1.000
*Bacteroidetes*	*Bacteroides paurosaccharolyticus*	0.004	0.014	0.019	0.014	1.000	1.000	1.000
*Bacteroidetes*	*Bacteroides vulgatus*	0.004	0.014	0.019	0.014	1.000	1.000	1.000
*Bacteroidetes*	*Parabacteroides goldsteinii*	0.007	0.051	0.027	0.016	1.000	1.000	1.000
*Deferribacteres*	*Mucispirillum schaedleri*	0.004	0.014	0.019	0.014	1.000	1.000	1.000
*Firmicutes*	*Absiella dolichum*	0.004	0.014	0.019	0.014	1.000	1.000	1.000
*Firmicutes*	*Acetatifactor muris*	0.004	0.014	0.019	0.014	1.000	1.000	1.000
*Firmicutes*	*Anaerotaenia torta*	0.004	0.014	0.019	0.014	1.000	1.000	1.000
*Firmicutes*	*Christensenella massiliensis*	0.004	0.014	0.019	0.014	1.000	1.000	1.000
*Firmicutes*	*Christensenella minuta*	0.004	0.014	0.019	0.014	1.000	1.000	1.000
*Firmicutes*	*Clostridium aerotolerans*	0.004	0.014	0.019	0.014	1.000	1.000	1.000
*Firmicutes*	*Clostridium asparagiforme*	0.004	0.014	0.019	0.014	1.000	1.000	1.000
*Firmicutes*	*Clostridium cocleatum*	0.004	0.014	0.019	0.014	1.000	1.000	1.000
*Firmicutes*	*Clostridium populeti*	0.004	0.014	0.019	0.014	1.000	1.000	1.000
*Firmicutes*	*Clostridium saccharolyticum*	0.004	0.014	0.019	0.014	1.000	1.000	1.000
*Firmicutes*	*Clostridium scindens*	0.004	0.014	0.019	0.014	1.000	1.000	1.000
*Firmicutes*	*Desulfitobacterium metallireducens*	0.004	0.014	0.019	0.014	1.000	1.000	1.000
*Firmicutes*	*Eisenbergiella massiliensis*	0.004	0.014	0.019	0.014	1.000	1.000	1.000
*Firmicutes*	*Eubacterium coprostanoligenes*	0.004	0.014	0.019	0.014	1.000	1.000	1.000
*Firmicutes*	*Eubacterium siraeum*	0.004	0.014	0.019	0.014	1.000	1.000	1.000
*Firmicutes*	*Falcatimonas natans*	0.004	0.014	0.019	0.014	1.000	1.000	1.000
*Firmicutes*	*Flintibacter butyricus*	0.004	0.014	0.019	0.014	1.000	1.000	1.000
*Firmicutes*	*Hungateiclostridium thermocellum*	0.004	0.014	0.019	0.014	1.000	1.000	1.000
*Firmicutes*	*Hydrogenoanaerobacterium saccharovorans*	0.004	0.014	0.019	0.014	1.000	1.000	1.000
*Firmicutes*	*Intestinimonas butyriciproducens*	0.004	0.014	0.019	0.014	1.000	1.000	1.000
*Firmicutes*	*Kineothri alysoides*	0.004	0.014	0.019	0.014	1.000	1.000	1.000
*Firmicutes*	*Lachnoclostridium pacaense*	0.004	0.014	0.019	0.014	1.000	1.000	1.000
*Firmicutes*	*Lactobacillus animalis*	0.004	0.014	0.019	0.014	1.000	1.000	1.000
*Firmicutes*	*Lactobacillus rogosae*	0.004	0.014	0.019	0.014	1.000	1.000	1.000
*Firmicutes*	*Muricomes intestini*	0.004	0.014	0.019	0.014	1.000	1.000	1.000
*Firmicutes*	*Neglecta timonensis*	0.004	0.014	0.019	0.014	1.000	1.000	1.000
*Firmicutes*	*Oscillibacter ruminantium*	0.004	0.014	0.019	0.014	1.000	1.000	1.000
*Firmicutes*	*Oscillibacter valericigenes*	0.004	0.014	0.019	0.014	1.000	1.000	1.000
*Firmicutes*	*Peptococcus simiae*	0.004	0.014	0.019	0.014	1.000	1.000	1.000
*Firmicutes*	*Pseudoflavonifractor phocaeensis*	0.004	0.014	0.019	0.014	1.000	1.000	1.000
*Firmicutes*	*Roseburia faecis*	0.004	0.014	0.019	0.014	1.000	1.000	1.000
*Firmicutes*	*Ruminococcus gnavus*	0.004	0.014	0.019	0.014	1.000	1.000	1.000
*Firmicutes*	*Vallitalea pronyensis*	0.004	0.014	0.019	0.014	1.000	1.000	1.000
*Proteobacteria*	*Hafnia alvei*	0.004	1.000	1.000	0.014	1.000	0.014	0.019
*Proteobacteria*	*Klebsiella variicola*	0.004	1.000	1.000	0.014	1.000	0.014	0.019

^1^ Mouse groups: T (Tim3VdIg-treated group), T/A (Tim3VdIg- and antibiotic-treated group), T/A/F (Tim3VdIg-, antibiotic-, and faecal bacteria-treated group), T/A/E (Tim3VdIg-, antibiotic-, and *E. hirae*-treated group) ^2^ Analysed by the Kruskal–Wallis test with Dunn’s multiple comparison.

**Table 3 microorganisms-08-01395-t003:** Bacterial species with significantly varied abundance between C57BL/6 and BALB/c mice treated with Tim3VdIg ^1^.

Phylum	Class	Species	*p* Value ^2^	More in
*Bacteroidetes*	*Bacteroidia*	*Bacteroides xylanolyticus*	0.01	C57BL/6
*Firmicutes*	*Clostridia*	*[Clostridium] indolis*	0.01	
*Firmicutes*	*Clostridia*	*Blautia producta*	0.011	
*Firmicutes*	*Clostridia*	*Caecibacterium sporoformans*	0.014	
*Firmicutes*	*Clostridia*	*Hungateiclostridium thermocellum*	0.01	
*Firmicutes*	*Clostridia*	*Pseudoflavonifractor phocaeensis*	0.01	
*Firmicutes*	*Erysipelotrichia*	*[Clostridium] cocleatum*	0.014	
*Firmicutes*	*Erysipelotrichia*	*Turicibacter sanguinis*	0.011	
*Proteobacteria*	*Gammaproteobacteria*	*Klebsiella variicola*	0.011	
*Proteobacteria*	*Gammaproteobacteria*	*Proteus alimentorum*	0.013	
*Bacteroidetes*	*Bacteroidia*	*Bacteroides caccae*	0.006	BALB/c
*Bacteroidetes*	*Bacteroidia*	*Bacteroides paurosaccharolyticus*	0.006	
*Bacteroidetes*	*Bacteroidia*	*Bacteroides vulgatus*	0.006	
*Bacteroidetes*	*Bacteroidia*	*Parabacteroides goldsteinii*	0.006	
*Deferribacteres*	*Deferribacteres*	*Mucispirillum schaedleri*	0.006	
*Firmicutes*	*Bacilli*	*Lactobacillus animalis*	0.006	
*Firmicutes*	*Bacilli*	*Lactobacillus rogosae*	0.006	
*Firmicutes*	*Clostridia*	*[Clostridium] viride*	0.006	
*Firmicutes*	*Clostridia*	*Anaerovorax odorimutans*	0.006	
*Firmicutes*	*Clostridia*	*Blautia faecis*	0.005	
*Firmicutes*	*Clostridia*	*Christensenella minuta*	0.006	
*Firmicutes*	*Clostridia*	*Flavonifractor plautii*	0.006	
*Firmicutes*	*Clostridia*	*Intestinimonas butyriciproducens*	0.009	
*Firmicutes*	*Clostridia*	*Peptococcus simiae*	0.006	
*Firmicutes*	*Clostridia*	*Vallitalea pronyensis*	0.006	
*Firmicutes*	*Erysipelotrichia*	*Absiella dolichum*	0.006	

^1^ The microbiome was analysed on day 14 after tumour challenge and treatment with Tim3VdIg five times, once every second day following tumour challenge. The number of samples was 6 for C57BL/6 and 4 for BALB/c mice. ^2^ Statistical significance of the fold difference of species abundance between BALB/c and C57BL/6 greater than 10 was analysed by the Wilcoxon rank sum test.

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
