# Peer review of "Modulation of the Gut Microbiota Alters the Tumour-Suppressive Efficacy of Tim-3 Pathway Blockade in a Bacterial Species- and Host Factor-Dependent Manner"

_microorganisms, 2020, doi:10.3390/microorganisms8091395_

Round 1

Reviewer 1 Report

Many thanks for inviting me to review this paper named "Modulation of the Gut Microbiota Alters the Tumor-Suppressive Efficacy of Tim-3 Pathway Blockade in a Bacterial Species- and Host Factor-Dependent Manner" by Bokyoung Lee et al. 

This work aims to dissect the role of microbiota modulation on tumor-suppressive Efficacy of Tim-3 Pathway Blockade (an immune check-point just like PD-1 or CTLA-4) in the animal model (mouse). The field is exciting for modern oncology and the Authors conducted a very detailed study not only delivering antibiotics but also restoring gut microbiota through oral gavage. Moreover, they went further highlighting the role of 2 single strains such as E. hirae and L. johnsonii.

Introduction, methods, and results are well explained; discussion is appropriate, limits (i.e low sample size) are recognized. 

Many compliments. 

Author Response

Dear Reviewer:

We thank you for your encouragement.

As suggested, English editing was carried out once more in order to improve language and style. 

Thank you again for your kind comments.

Sincerely,

Sun Park

Department of Microbiology, Ajou University School of Medicine;

Department of Biomedical Sciences, The Graduate School, Ajou University;

Youngtongku Wonchondong San 5, Suwon 442-749, The Republic of Korea

Tel.: +82-31-219-5070

Reviewer 2 Report

The manuscript entitled “Modulation of the Gut Microbiota Alters the Tumor-Suppressive Efficacy of Tim-3 Pathway Blockade in a Bacterial Species- and Host Factor-Dependent Manner” is an intrigue original study which endorse the role of microbioma in the enhancing activity of immunocheckpoint inhibitors in cancer.

This manuscript is well written and balance. The data presented seem clear and the experiments repeatable, so far it should be published after some minor revisions: 

  • English should be revised
  • Abbreviations should be checked (for example in Introduction: TH1; NK; PD1…)
  • INTRODUCTION: Authors should describe better the role of microbiota as a strategy to improve cancer immune checkpoint inhibitors efficacy that is the clear clinical role that this study reflects (Authors can refer to Strategies to Improve Cancer Immune Checkpoint Inhibitors Efficacy, Other Than Abscopal Effect: A Systematic Review. Cancers (Basel). 2019;11(4):539)
  • INTRODUCTION: At the end of this section, Authors wrote: we examined whether the gut microbiota modulation influences the efficacy of Tim-3 blockade in mouse tumor models. I believe that the aim of this study is more complex and should better addressed by the Authors.
  • METHODS: Authors should specify CT-26 cell line
  • DISCUSSION: Authors should justify the reason of the choice of the “production and purification” of Tim3VdIg protein. Why didn’t they bought commercial formulations?

Author Response

Dear Reviewer:

We thank you for your thoughtful suggestions and insights. The manuscript has benefited from these insightful suggestions. 

The manuscript has been rechecked and the necessary changes have been made in accordance with your suggestions. The revisions made in the manuscript to address your concerns have been shown in red-colored font. 

The responses to all comments have been prepared and attached herewith.

This manuscript is well written and balance. The data presented seem clear and the experiments repeatable, so far it should be published after some minor revisions: 

  • English should be revised

Response:

As suggested, English editing was carried out once more in order to improve language and style.

  • Abbreviations should be checked (for example in Introduction: TH1; NK; PD1…)

Response:

             Abbreviations were defined at their first appearance in the main text.

  • INTRODUCTION: Authors should describe better the role of microbiota as a strategy to improve cancer immune checkpoint inhibitors efficacy that is the clear clinical role that this study reflects (Authors can refer to Strategies to Improve Cancer Immune Checkpoint Inhibitors Efficacy, Other Than Abscopal Effect: A Systematic Review. Cancers (Basel). 2019;11(4):539)

Response:

Thank you for your recommendation of an appropriate reference. Accordingly, the following sentences were added to the Introduction.

Microbiota modulation has thus been suggested as a strategy to improve the efficacy of immune checkpoint inhibitors [27]. However, whether microbiota modulation can also enhance immune checkpoint inhibition efficacy in patients receiving antibiotics remains unclear.

INTRODUCTION: At the end of this section, Authors wrote: we examined whether the gut microbiota modulation influences the efficacy of Tim-3 blockade in mouse tumor models. I believe that the aim of this study is more complex and should better addressed by the Authors.

Response:

Thank you for the critical suggestion. The aim of the study was revised as follows.

In this study, we examined whether gut microbiota modulation influences the efficacy of Tim-3 blockade in mouse tumour models. In particular, the efficacy of gut microbiota modulation for enhancing the tumour-suppressive effects of Tim-3 blockade was evaluated in mice receiving antibiotics. We observed that oral gavage of bacteria altered the gut microbiota of mice despite continuous antibiotic administration. Further, the anti-tumour efficacy of Tim-3 blockade depended on the particular bacterial species administered through oral gavage and the mouse strain.

  • METHODS: Authors should specify CT-26 cell line

Response:

Based on your feedback, the relevant sentence in the methods section was edited as “CT-26 BALB/c colon carcinoma cells (provided by Dr. Kwon, Ajou University) [28] were cultured...“

  • DISCUSSION: Authors should justify the reason of the choice of the “production and purification” of Tim3VdIg protein. Why didn’t they bought commercial formulations?

Response:

According to your suggestion, the following description was added to the Discussion section.

Tim-3 pathway blockade is a promising form of cancer immunotherapy. Thus, the identification of factors that affect its anti-tumour efficacy is of great significance. In the current study, the anti-tumour effect of Tim3VdIg, a novel Tim-3-blocking molecule, was demonstrated, and the influence of gut microbiota on the therapeutic effects of Tim3VdIg in mice was analysed. …

The anti-tumour effect of Tim-3 blockade has been reported using anti-Tim-3 antibodies or the monomeric form of the Tim-3-Ig fusion protein [14, 19]. The Tim3VdIg used in the current study is a dimer form of the Tim-3V domain and Ig fusion protein and is not commercially available. While it was expected that Tim3VdIg may bind to Tim-3 ligands more stably than that by the Tim3VmIg monomer, the tumour-suppressive effect of Tim3VdIg was comparable to that of Tim3VmIg. However, the production yield of Tim3VdIg was higher than that of Tim3VmIg in the preliminary experiment (data not shown). Tim3VdIg injection inhibited tumour growth in both C57BL/6 and BALB/c mouse strains previously challenged with melanoma and colon carcinoma cells, respectively, indicative of its anti-tumour effect in hosts with different genetic background and different tumour type. 

Sincerely,

Sun Park

Department of Microbiology, Ajou University School of Medicine;

Department of Biomedical Sciences, The Graduate School, Ajou University;

Youngtongku Wonchondong San 5, Suwon 442-749, The Republic of Korea

Tel.: +82-31-219-5070
